# Performance Analysis of a Grid-Connected Rooftop Solar Photovoltaic System

**Alper Nabi Akpolat** [1,*] , **Erkan Dursun** [1] , **Ahmet Emin Kuzucuoğlu** [1] , **Yongheng Yang** [2,*] , **Frede Blaabjerg** [2] **and Ahmet Fevzi Baba** [1]

1  Department of Electrical-Electronics Engineering, Marmara University, 34722 Istanbul, Turkey
2  Department of Energy Technology, Aalborg University, 9220 Aalborg, Denmark
*  Correspondence: alper.nabi@marmara.edu.tr (A.N.A.); yoy@et.aau.dk (Y.Y.);
   Tel.: +90-535-3377520 (A.N.A.); +45-9940-9766 (Y.Y.)

**Abstract:** Turkey is among the countries largely dependent on energy import. This dependency has increased interest in new and alternative energy sources. Installation of rooftop solar photovoltaic systems (RSPSs) in Turkey is increasing continuously regarding geographical and meteorological conditions. This paper presents an insight into the potential situation for Turkey and a simulation study for the RSPS designing and calculation for the faculty building at Marmara University in Istanbul. This simulation study demonstrates that 84.75-kWp grid-connected RSPS can produce remarkable power. The system is performed in detail with the PV*SOL software (Premium 2017 R8-Test Version, Valentin Software GmbH, Berlin, Germany). Detailed financial and performance analysis of the grid-connected RSPS for faculty building with various parameters is also carried out in this study. According to the simulation results, the system supplies 13.2% of the faculty buildings' annual electrical energy consumption. The annual savings value of faculty buildings' electrical consumption is approximately 90,298 kWh energy which costs roughly $7296. A photovoltaic (PV) system installation for the faculty building, which has considerable potential for solar energy and sunshine duration, is indispensable for clean energy requirements and was supported by the simulation results. This paper can be considered to be a basic feasibility study prior to moving on to the implementation project.

**Keywords:** faculty building; performance analysis; rooftop solar photovoltaic

## 1. Introduction

Solar energy is the primary source of energy that affects physical formations in the space and atmosphere system. The solar energy that falls on the earth every year is about 160 times higher than the fossil fuel reserves that have been determined up to now on Earth. Moreover, energy production from fossil, nuclear, and hydroelectric plants produced in a year is 15,000 times less than solar energy [1–3]. Just as the energy can come into existence like chemical, potential, kinetic in the natural sciences, it may take place in nature in different forms such as the sun and the wind. However, electrical energy is generally obtained by conversion from different energy forms, and the energy sources that make up the first step of this process are mostly fossil fuels. Fossil fuels account for about 80% of the world's total energy production. The burning of fossil fuels releases harmful gasses such as $CO_2$, $SO_2$, $CO$, $NO_2$, and $NO_3$. The release of these gasses triggers irregular climate changes. For all these reasons, the demand for environmentally friendly, renewable energy systems has increased over the last decade [4]. In our modern world, there is a parallelism between the level of development and the economic situation of society and energy production/consumption. Energy costs create an undeniable impact on the economic well-being of a country [5]. Across the world, population and energy consumption

in parallel are dramatically ascending after the industrial revolution [6]. At any cost, the world has transition immediately to green energy, of which renewable energy sources (RESs) are preferred.

The two main resources for very large scale renewable energy (RE) harvesting are the wind and solar resources. The wind resource may show variability and be limited, but the harvesting of solar resources is easier and more common than the wind.

Therefore, solar energy is a suitable technology for both small and large scale applications. It is clean energy according to the principle of sustainability. In particular, solar energy is the fastest-growing energy technology in the world. Solar energy includes the two main way as photovoltaic (PV) and concentrating solar power (CSP). PV is unquestionably more applicable than CSP [7,8].

Therefore, generating energy with solar PV is the most trending application in terms of using RESs across the world. Recently, in developing countries across the world, decisions are being debated and a transitioning to the dependence from fossil fuels to RES. Among renewable energy, especially the PV systems play a vital role in this transition for PV applications.

In the literature, in terms of solar energy is examined, it was observed that studies on the rooftop solar photovoltaic applications especially in faculty buildings are not very extensive. In regard to installation integrated PV systems, Goia and Gustavsen present the analysis of the performance of the semi-integrated PV system of the ZEB Living Laboratory at Norwegian University of Science and Technology (NTNU). The payback ratio value of the system was calculated for time steps of five minutes based on electric energy meters' readings every 30 s [9]. Aelenei et al. investigated the electricity generation performance of 12-kWp installed power on the south facade a building integrated photovoltaic (BIPV) and an additional 12-kWp PV roof system in a nearby car park facility for office buildings at the National Laboratory of Energy and Geology (LNEG), in Portugal [10]. Dondariya et al. [11] focused on feasibility analysis of grid-connected rooftop solar photovoltaic system (RSPS) at Ujjain Engineering College, in India. The study presents a comparison of various simulation software, including PV*SOL, PVGIS, SolarGIS, and SISIFO for RSPS. Sayegh et al. [12] designed an experimental model of a Building Integrated Photovoltaic/Thermal air collector (BIPV/T) and it is installed on the roof of the Faculty of Mechanical Engineering at the University of Aleppo. Allouhi et al. evaluated the levelized cost of electricity, payback time and annual avoided $CO_2$ emissions of PV systems installed on the roof of an administrative building [13]. A paper by Patarau et al. presented the techno-economic analysis and optimization in a hybrid renewable energy system (HRES) using the PV*SOL software, consisting of solar panels, geothermal batteries, and a biomass generator in a greenhouse in Romania [14]. Another BIPV study was conducted in Indonesia [15]. Across the world, many academics have been studying PV systems. Their partial shading effects, power profiles under different conditions, behaviors under real experimental test platforms, integration with installed power electronics, etc. are widespread research areas of the RSPSs [16–18]. Moreover, the various studies by the PV*SOL have been presented at different countries. In India, a solar PV system was simulated for a hostel building [19]. In the study by Işık et al. [20], analytical and the PV*SOL software analysis of the RSPS were carried out and the results were compared.

Similarly, this study is to aim to investigate the applicability and feasibility of grid-connected rooftop PV systems in the faculty buildings in detail, by means of the necessary simulation software. Before projecting PV systems based on the distributed generation (DG), it is clear that the relevant system can be applied by making different PV system simulation tools [21]. In this study, the PV*SOL, energy production performance, economic analysis, carbon emission, and other simulation outputs/results of the system are demonstrated. In the analysis of the pre-installation of solar PV systems, it is proposed that the basic calculations are not adequate, and sophisticated software support should be focused on.

The contribution of this paper is to design and analysis of the simulation of an RSPS to conjecture the power for improving the system performance as well as to help in integrating into the power grid system of the faculty building. The outputs of the designed simulation may be a guide for similar applications and helpful for researchers and engineers so as to evaluate the establishment, behavior,

and operation of a rooftop PV conversion system. This simulation results are a reference study for university campuses and municipalities on transitioning to smart, green, and clean energy building concept that refers to "Smart Grid". Accordingly, the organization of the paper is as follows. Section 2 describes renewable energy status of Turkey. The system sizing and relevant equations is referred to in Section 3. The simulation results and analysis are presented in Section 4. Finally, discussion and concluding remarks are introduced in Section 5.

## 2. Renewable Energy Status of Turkey

Turkey has substantial renewable energy potential, including solar, mainly as a result of its geographic location. Primary energy production (thousand tonnes of oil equivalents-ktoe) by years in Turkey is shown in Figure 1. The gradual incremental production of renewable energies in the country is clearly seen. Tone of oil equivalent (toe) is a unit of energy, defined as the amount of energy released burning one tonne (1000 Kilograms) of crude oil. One thousand tonnes of oil equivalent (ktoe) is equal to 11,630,000 kWh energy.

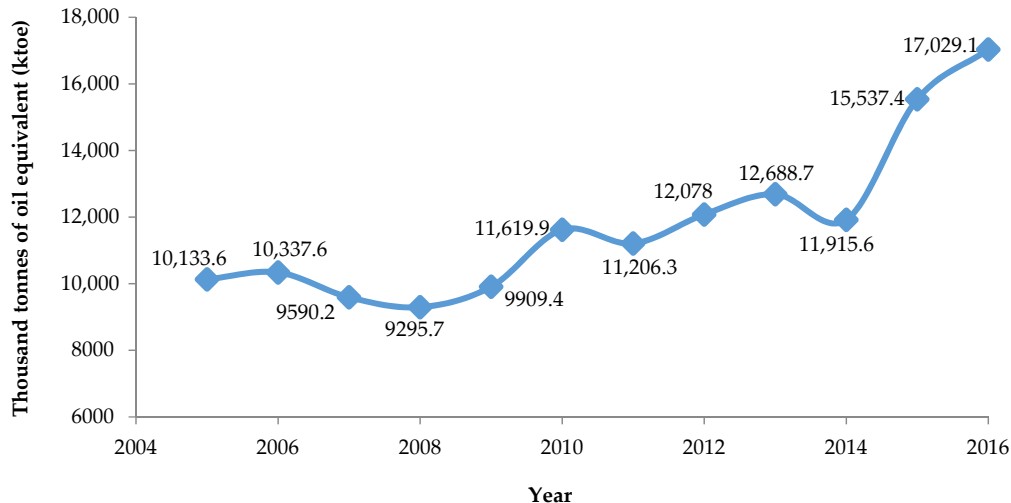

**Figure 1.** The graphic of primary production data from [22] by renewable energies in Turkey.

The advantage of the potential will lower the country's dependence on imported fossil fuels as well as lessen greenhouse gas emissions. The values of Turkey's Solar Generation Energy Potential Atlas (GEPA) alleges to have been an annual sunshine time of 2.741 h (7.5 h per day) and the total incoming solar energy per square meter is 1527 kWh per year with daily average of 4.18 kWh [22]. In addition, the Turkish government has intended to reach a goal of at least 30% (or 127.3 TWh) of the total electricity generation from renewable energy sources by 2023. Moreover, the government aims to expand its installed solar power more than 2 GW from 2019 to 2023.

The installed solar power generation in Turkey has grown in the last few years with solar capacity changed from 40 MW to 832.5 MW from 2014 to 2016 in Table 1 and has reached about 3421 MW at the end of 2017 [23].

**Table 1.** Primary Energy Resources Data Obtained from [24] in Turkey.

| MW | Coal | Liquid Fuel | Natural Gas | RE/Waste Heat | Multi-Fuel | Hydraulic | Geothermal | Wind | Solar | Total |
|---|---|---|---|---|---|---|---|---|---|---|
| 2006 | 10.2 | 2396.5 | 11,462.2 | 41.3 | 3323.4 | 13,062.7 | 23.0 | 59.0 | - | 40,564.8 |
| % | 25.14 | 5.91 | 28.26 | 0.10 | 8.19 | 32.20 | 0.06 | 0.15 | - | 100.00 |
| 2016 | 17,355.3 | 445.3 | 19,563.6 | 496.4 | 6551.0 | 26,681.1 | 820.9 | 5751.3 | 832.5 | 78,497.4 |
| % | 22.11 | 0.57 | 24.92 | 0.63 | 8.35 | 33.99 | 1.05 | 7.33 | 1.06 | 100.00 |

According to the International Energy Agency (IEA) installed photovoltaic (PV) power in Turkey is of a slight rise that can be seen in Table 2. It can be shown that hopeful incremental is expected to grow gradually in the following years.

**Table 2.** Installed Photovoltaic (PV) Power Data from [25] in Turkey (2017).

| Month | Total Installed (MW) | | |
|---|---|---|---|
| | Licensed | Unlicensed | Total |
| January | 12.90 | 847.70 | 860.60 |
| February | 12.90 | 886.00 | 898.90 |
| March | 12.90 | 1041.20 | 1054.10 |
| April | 12.90 | 1090.80 | 1103.70 |
| May | 12.90 | 1152.80 | 1165.70 |
| June | 12.90 | 1349.70 | 1362.60 |
| July | 12.90 | 1460.11 | 1473.01 |
| August | 12.90 | 1566.44 | 1579.34 |
| September | 12.90 | 1752.92 | 1765.82 |
| October | 13.90 | 2046.20 | 2060.10 |
| November | 13.90 | 2231.80 | 2245.70 |
| December | 17.90 | 3402.80 | 3420.70 |

The installation of PV system close to energy consumption points will reduce energy transmission and distribution losses. For this reason, the installation of RSPSs needs to be widespread, especially in urban areas where energy consumption is high. As well as on the RSPSs in buildings, the use of PV systems in the facade cladding systems may be preferred. Potential of the physical space, area of use, and technical potential of RSPS in Turkey is shown in Table 3.

**Table 3.** According to Rooftop Solar Market Assessment Summary Note Data from [23], the Potential of The Rooftop Solar Photovoltaic System (RSPS) in Turkey.

| Building Type | # of Buildings (in thousands) | Base Area (million m$^2$) | Weighted Average Usable Area | Usable Area (million m$^2$) | RSPS Technical Potential (GW) |
|---|---|---|---|---|---|
| Residential | 8230 | 1269 | 47% | 596 | 23.2 |
| Commercial/ Industrial | 950 | 875 | 57% | 499 | 21.5 |
| Public | 69 | 93 | 45% | 42 | 2.1 |
| Total | 9248 | 2237 | - | 1.137 | 46.8 |

Estimated annual RSPS growth in Turkey is shown in Figure 2.

The concept of energy-efficient buildings has emerged with the transformation of energy-consuming structures into energy-producing via renewable sources [26]. University campuses are active areas with student numbers, laboratories, classes, and offices. Energy-efficient buildings need to have a transition to green building without depriving of the quality of education. Effective use of energy in university buildings is indispensable not only in terms of sustainable energy but also regarding continuing the trends, culture, and education in the future professions of the persons who are studying in these buildings.

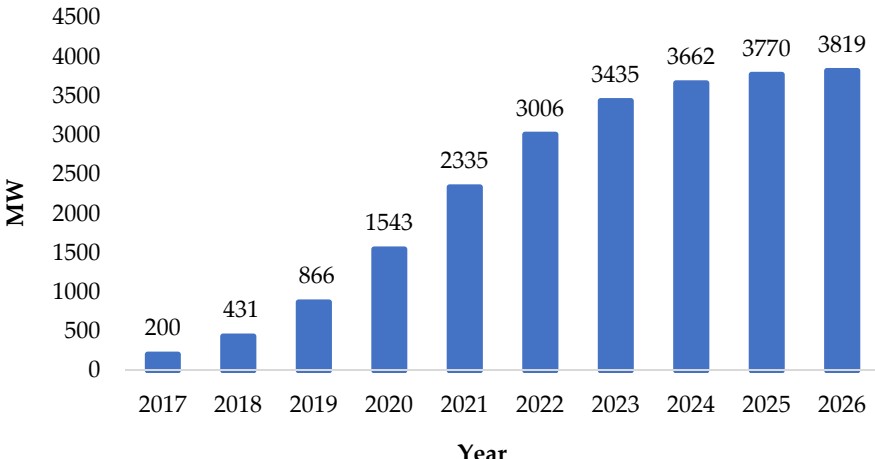

**Figure 2.** The column chart of estimated annual RSPS growth data exist in [23] for Turkey.

## 3. System Sizing and Methodology

Goztepe campus of Marmara University has an area of 153,394 m$^2$ (153.3 acres) with six faculties, four schools, and five institutes. The Goztepe campus is located in a flat land in Kadıkoy district of Istanbul. The geographic coordinates of the campus are 40°59' North Latitude, 29°3' East Longitude [27]. Figure 3 shows the climate data are examined for the current location in this study. The annual average temperature is 14.6 °C, and the average solar irradiance is 4.19 kW/m$^2$ per day for the campus. These values are precisely notable for installing a PV system.

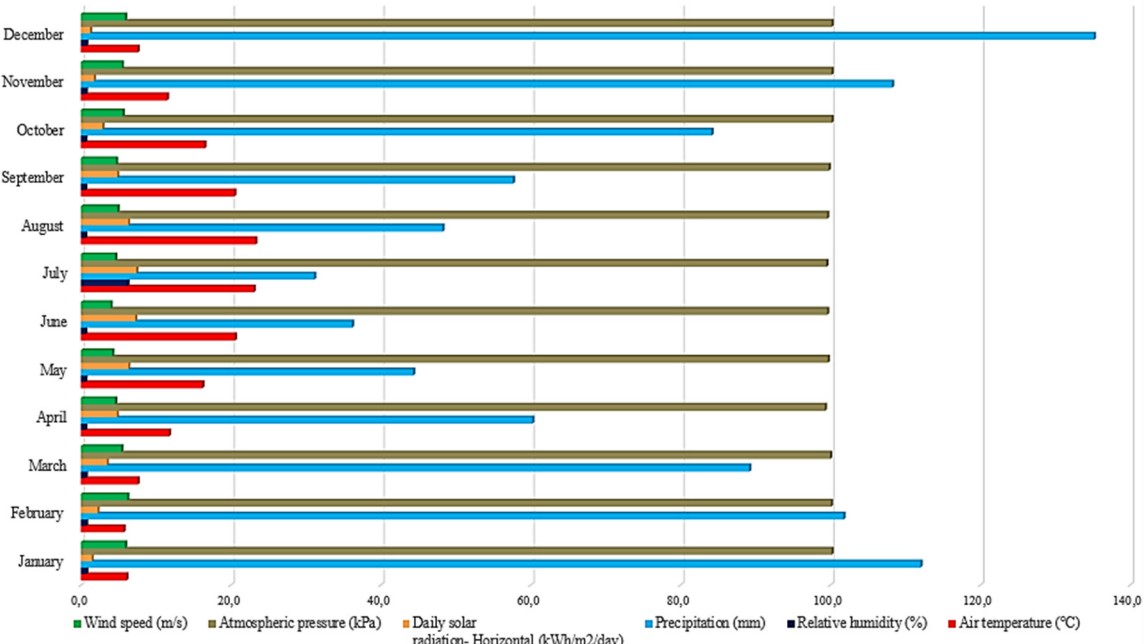

**Figure 3.** The drawn chart of climate data in [28] for the current location.

The annual global solar irradiation potential of Istanbul is about 1500 kWh/m$^2$, which is a notable value across the world as seen in following Figure 4.

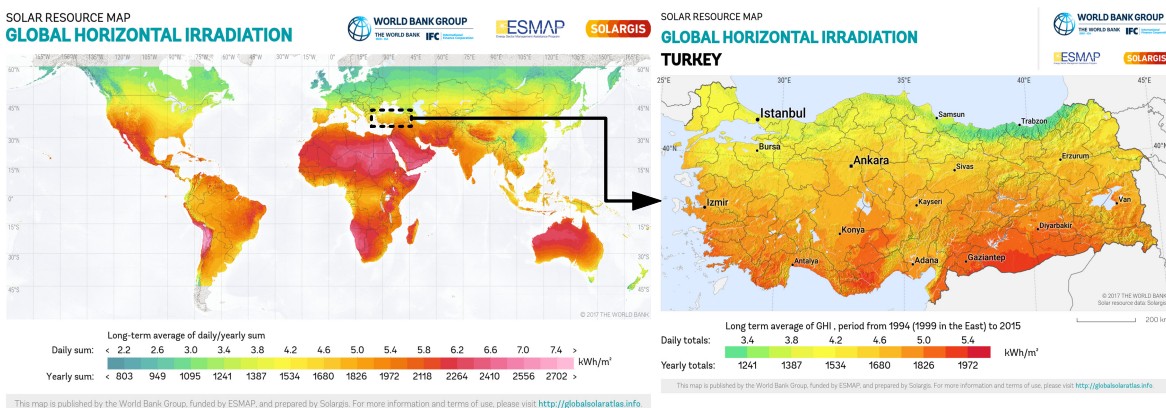

**Figure 4.** Global horizontal irradiation maps that exist in [29] for the Earth and Turkey.

Five transformers and 28 conductor lines provide the electrical energy in the campus of the university, which is positioned according to the needs. The existing energy infrastructure comes from a conventional electricity grid lacking RESs. These 28 conductor lines, separated from the four main feeders, are scattered throughout the campus. Instead of a meter connected to a single line that shows the energy consumption of the entire campus, there are four meters connected to four main feeders. Two of the meters are in the Rectorate, one is in the Faculty of Education, and the other is in the Faculty of Technology. Table 4 shows the installed capacities, annual average energy consumption and yearly average costs of these electricity meters, which are connected to four main feeder points.

**Table 4.** Electrical Energy Features in the Campus (2017).

| Feeder Location | Energy Consumption Table | | |
|---|---|---|---|
| | Installed Power (kW) | Annual Average Energy Consumption (kWh) | Annual Average Energy Costs (Turkish Lira-₺) |
| Rectorate 1 | 1000.00 | 2,428,848.06 | 910,816.87 |
| Rectorate 2 | 116.66 | 332,925.08 | 124,713.73 |
| Faculty of Education | 833.33 | 430,864.91 | 161,401.99 |
| Faculty of Technology | 1260.00 | 683,948.20 | 256,206.99 |
| Annual Total of Campus | 3209.99 | 3,876,586.26 | 1,452,169.21 |

### 3.1. PV System Components

Annual energy production potential of the PV system is based on solar panels, inverter, battery bank, solar cable, meteorological and geographical conditions, and physical characteristics. These variables affect the PV system results. Depending on the application, the solar cell modules are used in conjunction with accumulators, inverters, battery chargers, and various electronic support circuits to form solar PV systems [30].

In these systems, a sufficient number of solar modules are used as an energy source. A battery group is usually included in the system for use when the sun is not enough to generate current or during the night, especially. The solar cells generate electricity during the day, store it in the battery group, and the energy required for the load is taken from the accumulator. The control unit (charge regulator) used to prevent the damage from overcharging and discharging, cuts off the current generated from the battery. In applications that AC power is required, DC voltage in the accumulator is converted to a sine wave of 220 V-50 Hz by adding an inverter to the system.

Similarly, various electronic circuits can be integrated into the system according to the scheme of the application [31,32]. Some systems have a maximum power point tracking (MPPT) specification via an inverter that allows solar batteries to operate at maximum power as shown in Figure 5.

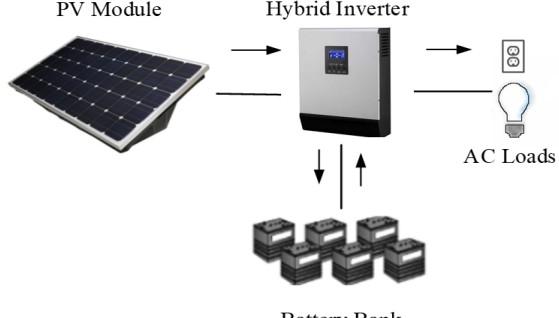

**Figure 5.** PV system components and configuration based on the concept.

According to the application, PV modules can be used together with accumulators, inverters, battery chargers, switching components, and various assistive electronic circuits to form a PV system. The solar energy may not only be applied economically in conditions where the fuel of generator is transported difficultly and expensively, but also where there is no electricity grid and where the generator is far from the settlements. In areas where the night consumption is quite low compared to the daytime consumption, systems connected directly to the grid (on-grid) may be preferred.

The methodology of the paper with PV*SOL software supports system designers in deciding on the PV system. The software evaluates the necessary data and calculates the solar yield. The orientation and inclination of the PV panel should be defined in advance. After choosing one of the three types of solar PV modules such as monocrystalline, polycrystalline, and thin film, the number of module size of the module array, the software automatically determines the location of the PV system. Alternatively, related to the system location, any place can be selected on the world map. PV*SOL software uses climate data from PV*SOL database.

The simulation contains quite a complicated process. Generally, the main steps of the simulation can be expressed such as selecting the location (the meteorological data), selecting PV modules and their planning, selecting inverters, cables or other devices requirements. After these complex definitions and detailed design, a detailed report can be printed out. With the climate data for the location and the characteristics selected for the solar PV system, the expected annual yield of the system can be calculated using a detailed hourly simulation. The simulation prefers the PV*SOL software calculation model equations on the Valentin Software, which we discuss in Section 3.3. so as to calculate the whole output results.

### 3.2. PV Manual Calculations

The amount of panel requirement for off-grid PV systems where the daily energy consumption, the average sunshine duration, and the power of the panel to be used are known, occurs in manual calculations. For instance, an application with the daily energy consumption of 1000 W and an average sunshine duration (*ASD*) of two hours. If the total daily consumption (*TDC*) is taken into account, it is required to calculate the power to acquire per hour as

$$P_{PH} = \frac{TDC}{ASD} = \frac{1000(Wh)}{2(h)} = 500W \tag{1}$$

If a panel with 250 Wp power is used, several of the panels (*n*) are calculated through dividing the power per hour ($P_{PH}$) by the panel power ($P_m$):

$$n = \frac{P_{PH}}{P_m} = \frac{500(W)}{250(W)} = 2 \tag{2}$$

According to this manual calculation, the system which has this consumption can be put forward to supply by two panels approximately. With more complex calculation, different parameters such

as highest/lowest temperature ($T_H/T_L$), inverter voltage/current ($V_{inv}/I_{inv}$), maximum power point voltage ($V_{mpp}$), open-circuit voltage ($V_{OC}$), short-circuit current ($I_{SC}$) of the solar panel etc. should be considered according to the subsequent equations. The produced solar panels demonstrate different characteristics under different temperature circumstances. For the different characteristics of the solar panels, the boundaries can be taken as a reference at the highest ($T_H$) and lowest ($T_L$) temperature. For the on-grid system, the maximum number of solar panels ($n_{max}$) that can be in series is

$$n_{\max} = \frac{V_{inv-\max_{(input)}}}{V_{OC-T_L}} \tag{3}$$

The minimum number of solar panels ($n_{min}$) and the maximum number of parallel string ($n_{parallelstring}$) that can be in series is

$$n_{\min} = \frac{V_{inv-\min_{(input)}}}{V_{mpp-T_H}} \tag{4}$$

With

$$n_{parallelstring} = \frac{I_{inv-\max_{(input)}}}{I_{SC-T_H}} \tag{5}$$

The highest ($T_H$) and lowest ($T_L$) temperature are vital to calculate $V_{OC}$ and $I_{SC}$ in the location where the solar panel will be used. Since both $V_{OC}$ and $I_{SC}$ reach their extreme values at the lowest and highest temperature respectively. Open-circuit voltage at the lowest temperature is $V_{OC-T_L}$ As

$$V_{OC-T_L} = V_{OC-25°C} \times (1 + \Delta t \times \alpha_P) \tag{6}$$

where $V_{OC-25°C}$ (V), the difference of the temperature $\Delta t = (25 - T_L)(°C)$, and $\alpha_p (\%/°C)$, is the open-circuit voltage under standard test condition (STC—1 kW/m$^2$ irradiance and at 25 °C temperature), temperature difference, and PV temperature coefficient of power, respectively. In addition, a maximum number of solar panels, in this case, are seen in Equation (3). The maximum voltage at the highest temperature is calculated according to the following equation as

$$V_{mpp-T_H} = V_{mpp-25°C} \times (1 + \Delta t \times \alpha_P) \tag{7}$$

where $V_{mpp-25°C}$ (V), is the voltage at the maximum power point under STC, and $\Delta t = (T_H - 25)$ (°C) respectively. In addition, a minimum number of solar panels, in this case, are seen in Equation (4). The maximum current $I_{TH}$(A) at the highest temperature is

$$I_{T_H} = I_{SC-25°C} \times (1 + \Delta t \times \alpha_P) \tag{8}$$

The number of parallel strings, in this case, are seen in Equation (5). As a result, of the calculations, many solar panels to be connected to an inverter ($n_{module}$) is procured by multiplying the maximum number of solar panels ($n_{max}$) and a maximum number of parallel string ($n_{parallelstring}$).

$$n_{module} = n_{\max} \times n_{parallelstring} \tag{9}$$

If the number of the module ($n_{module}$) and panel power ($P_m$ (W)) are multiplied, DC power ($P_{DC}$ (W)) connected to an inverter can be obtained as follows:

$$P_{DC} = n_{module} \times P_m \tag{10}$$

The system performance ratio (*SPR*) is an indication of the energy losses within the system that takes place in comparison with the nominal output of the system. The value of the *SPR*, regarding unit electric energy generated for the amount of unit radiation. That may change depending on generated total energy from PV plant (*Et* (kWh)), area per solar panel (*A* (m$^2$)), number of panel (*n*), the annual

average solar radiation on tilted panels ($H$ (kWh/m$^2$)) shadings not included, and efficiency ($\eta$ (%)). The value of *SPR* can be expressed as follows:

$$SPR = \left( \frac{Et}{A \times n \times H \times \eta} \right) \tag{11}$$

We selected an AXIblack solar panel and a Growatt inverter in Table 5, to perform manual calculations show that 96 panels can connect up to per inverter. With these 96 panels, it revealed that a total installed DC power of 24 kW could be obtained via an inverter.

According to this simulation results will be addressed in the following sections,

$Et$ = 90,298 kWh, $n$ = 339 panels, $A$ = 1.643 m$^2$, $H$ = 1527 kWh/m$^2$, $\eta$ = 15.37% with these values *SPR* is calculated as 69.04%. Except for that a new *SPR* value will be calculated in the simulation.

### 3.3. PV Software Calculations

The PV*SOL program compiles the algorithms that some bases of calculations which contain for the following topic areas such as radiation processor, temperature parameters, PV power outputs, inverter specifications, output losses, evaluation parameters, economic efficiency calculations, design recommendations on the background [33]. The simulation prefers the PV*SOL software calculation model equations on the Valentin Software model. For the radiation processor, in the climate data obtained, the radiation plane is given horizontally regarding watts for the square of the active solar surface (horizontal radiation).

Based on the presumption that the modules are compiled in the maximum power point (MPP) operation. The temperature dependence of the curve namely module efficiency at various module temperatures is calculated from the following formula at 25 °C ($\eta_{PV(mpp)}$ ($G$, *TModule* = 25 °C)) and the output temperature coefficient $d\eta dT$ as follows [33]

$$n_{PV(mpp)} = n_{PV(mpp)}(G, T_{Module} = 25°C) \times [1 + \Delta T \times d\eta dT] \tag{12}$$

The inverter has functions such that Direct Current (DC) production of the PV modules is transformed into Alternating Current (AC). The conversion from DC to AC give rise to losses. At this point, the AC power ($P_{AC}$) of the inverter can be obtained by multiplying DC power ($P_{DC}$), efficiency at nominal power ($\eta_{Nenn}$), and relative efficiency ($\eta_{rel}$) as

$$P_{AC} = P_{DC} \times \eta_{Nenn} \times \eta_{rel} \tag{13}$$

For the temperature of the PV module, the value of the constant $k$ depends on the PV module type and a manageable parameter at maximum irradiance ($G_{STC}$ = 1000 W/m$^2$) compared with the outdoor temperature $T_a$ [34]. In this model, the module temperature ($T_{Module}$) has dependence on the radiation $G$ can be calculated as

$$T_{Module} = T_a + k \times \frac{G}{G_{STC}} \tag{14}$$

The PV panel converts the solar power into the electrical power dependent on the temperature and its heat capacity. Regarding absorbed power ($Q_G$); variables are absorption coefficient ($\alpha$), module area ($A_{Module}$), for radiated thermal output power ($Q_S$); variables are installation factor ($f_E$), emission coefficient ($\varepsilon$), Stefan-Boltzmann constant ($\sigma$), lastly for convection power ($Q_K$); variables are wind speed ($V_w$) and characteristic overcurrent length ($I_{char}$). By means of these variables, the powers can be calculated as follows [33]

$$Q_G = \alpha \times G \times A_{Module} \tag{15}$$

$$Q_S = f_E \times \varepsilon \times A_{Module} \times \sigma \times (T_{Module}^4 - T_a^4) \tag{16}$$

$$Q_K = f(A_{Module}, T_{Module}, T_a, V_W, I_{char}) \tag{17}$$

Also, these powers are connected with module dimension ($m_{Module}$), the heat capacity of module ($C_{Module}$), rate of the module temperature change ($dT_{Module}/dt$), and electrical power of the output ($P_{el}$) are given as

$$m_{Module} \times C_{Module} \times \frac{dT_{Module}}{dt} + P_{el} = Q_G - Q_S - Q_K \tag{18}$$

Additionally, the base calculation of the cable resistance ($R$) regarding the power of cabling losses can be also calculated from the variables of following formulas such as specific resistance of the material ($\sigma$), cable length ($l$), and cable cross-section ($A$).

$$R = \sigma \times \frac{l}{A} \tag{19}$$

With

$$P_R = U_R \times I = I^2 \times R \tag{20}$$

The following parameters are included in the software algorithm. The solar fraction ($SF$) depends on generated solar energy ($E_{PV(use)}$) and electricity requirement of consumer ($E_{Last}$) as

$$SF = \frac{E_{PV(use)}}{E_{Last}} \tag{21}$$

The specific annual yield ($SAY$ (kWh/kWp)) depends on $E_{PV(use)}$ and the installed PV array output power ($P_{nom}$), can be calculated as

$$SAY = \frac{E_{PV(use)}}{P_{nom}} \tag{22}$$

The $SPR$ is related to $E_{PV(use)}$, solar energy to PV surface ($E_{in}$), and the energy losses with the module efficiency at STC ($\eta_{STC}$), is calculated as follows

$$SPR = \frac{E_{PV(use)}}{E_{in}} \times \eta_{STC} \tag{23}$$

The system efficiency ($SE$) is of a measure of the conversion of the total energy irradiated to the array surface, can be expressed as

$$SE = \frac{E_{PV(USE)}}{E_{in}} \tag{24}$$

Moreover, the PV*SOL software has sophisticated calculation tools and uses the Net Present Value ($NPV$) method for economic calculations. The cash value of costs ($CVc$) of a price-dynamic payment series over lifetime ($T$) on the way Z, Z*r, Z*r²..., q is simple interest factor (1.08 at 8% simple interest), r is price change factor (1.1 at 10% price change), according to VDI 6025 standard [33,35] is given as

$$[CVc] = Z \times b(T,q,r) \tag{25}$$

With

$$b(T,q,r) = \begin{cases} 1 - (r/q)^T \rightarrow (r \neq q) \\ T/q \rightarrow (r = q) \end{cases} \tag{26}$$

If a constant series of payment is r = 1, annuity factor and T as year, on the way are given as follows

$$\alpha(q,T) = 1/b(T,q,r), for \rightarrow r = 1 \tag{27}$$

with

$$Z = [CVc] \times a(q,T) \tag{28}$$

To calculate the net present value of investments ($NPV_i$) it needs to be a summing of the cash value of the lifetime ($CV_l$), investments ($INVS$), and additional subsidies ($SUBS$). The equation for the annual electricity generation costs ($AEGC$) can be calculated by dividing payment series over lifetime on the way ($Z$) into the annual electricity generation ($AEG$) as follows [33]:

$$NPV_i = \sum (CV_l + INVS) + SUBS \tag{29}$$

$$AEGC = \frac{Z}{AEG} \tag{30}$$

## 4. System Designing and Simulation Results

It is necessary to reveal the consumption data shown in Figure 6 to analyze the results for the relevant system through the PV*SOL simulations. To submit the different system parameters, input data for the simulation is needed.

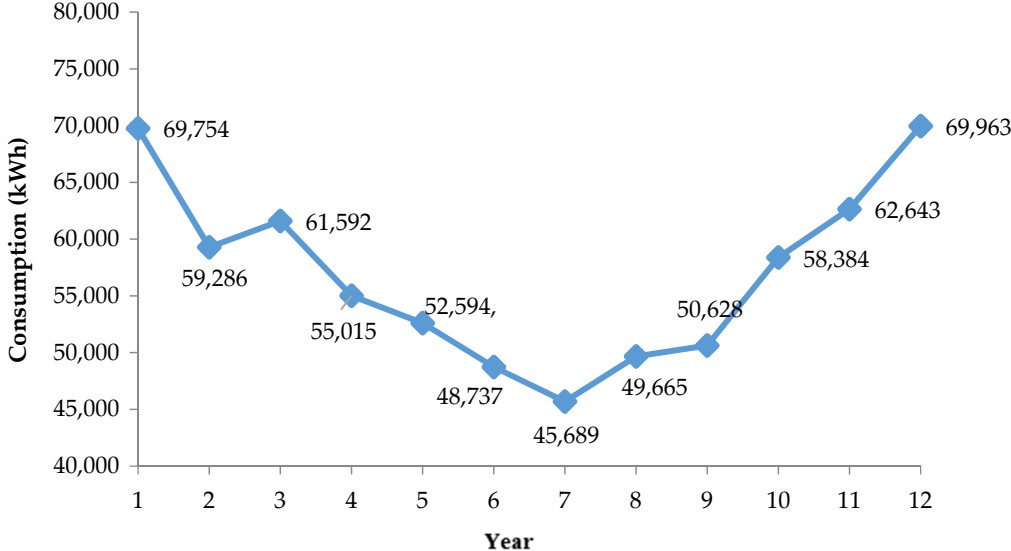

**Figure 6.** The annual average energy consumption of the faculty building as to the 2017 year.

Lots of data are loaded into the system after the step of determining the main data, the climate data, the grid and the design type for the system to be simulated. A three-dimensional (3D) model and a near-realistic outline are drawn in the software.

The PV panel to be installed on this sketch is placed in the desired type and angle, making the installation to proceed to the next steps. It follows that the area or building is scanned and designed with real features to establish the system.

The interface of the software entering inputs data of the system is shown in Figure 7. Before the simulation begins, the necessary data inputs which are system type, climate, location, infrastructure, and grid specifications, can be performed using this interface.

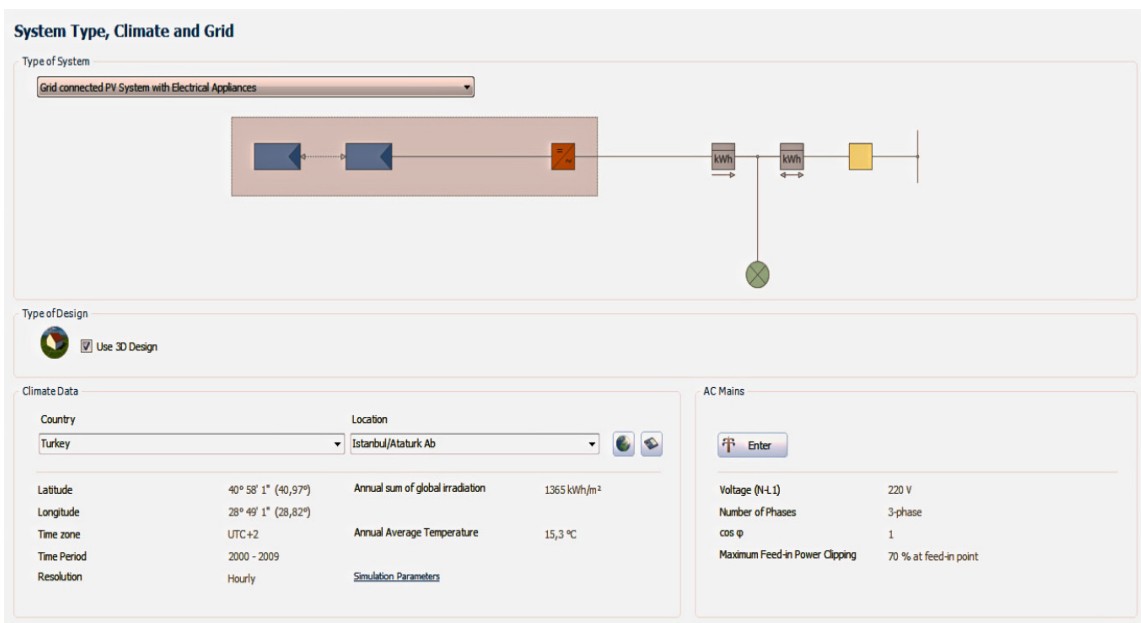

**Figure 7.** Main system type and inputs.

To place PV panels on the rooftop of the faculty building as the planned area of design, it is necessary to take out the actual three-dimensional model by taking advantage of the position and physical characteristics of the building. The model by actual characteristics is shown in Figure 8.

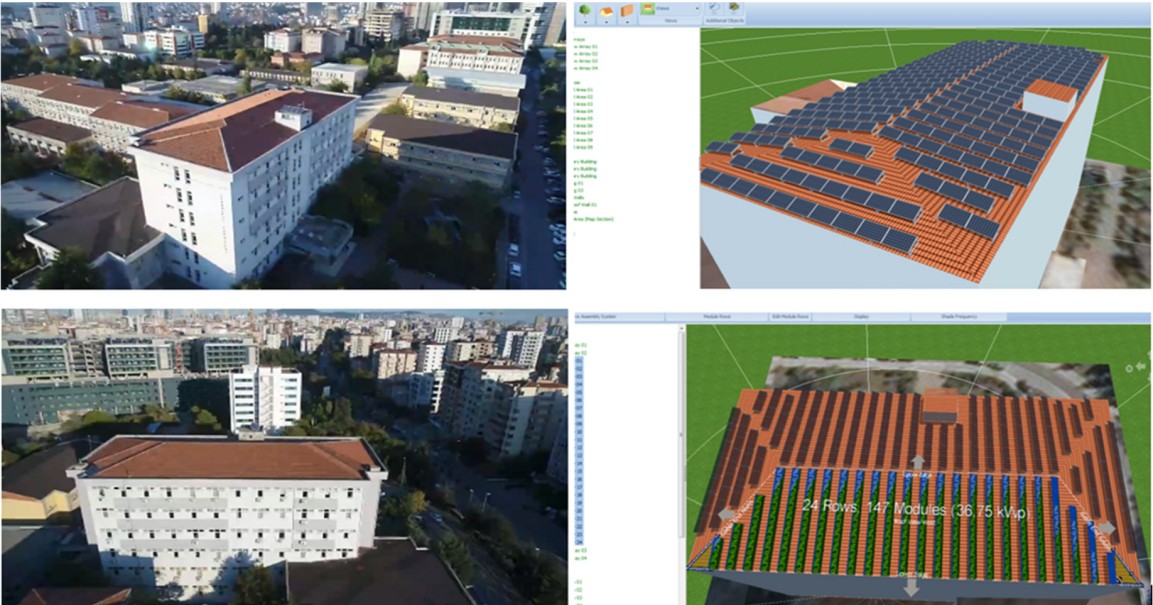

**Figure 8.** Faculty building and its 3D model mounted-roof PV panels.

In this respect, in searching for the installation type of PV panels for Turkey due to geographical and physical conditions, the mounted-roof installation type appears to have been generally preferred for these systems [36]. Thus, we opted for mounted-roof installation type after the 3D model is drawn in the simulation. The faculty building has a roughly rectangular shape in the north-south direction. The roof of the building is structurally built like a cone, which has four different planes due to its climatic properties.

This system is positioned just about 1170 m$^2$ of building roof surface area for PV modules, but PV generator surface area is 551.5 m$^2$. Therefore, while the roof surface is designed, different

areas are defined for each of the east-west and north-south four facades. These four subsystems, east-west-north-south groups, which will be positioned according to these areas, unite and result in the primary system. The selected panel-type is 60 cell/monocrystalline high-performance PV module that has 8.80 A short-circuit current ($I_{SC}$), and 37.98 V open-circuit voltage ($V_{OC}$) is approximately 250 Wp in capacity. For all the calculations, selected solar panel (at STC, irradiance 1000 W/m$^2$, spectrum AM 1.5 at a cell temperature of 25 °C) and inverter specifications for simulation are shown in Table 5.

**Table 5.** Preferred Solar Panel and Inverter Specifications.

| Solar Panel AXIblack AC-250 M/156-60 S | | Growatt 40000TL3-Inverter | |
|---|---|---|---|
| Nominal Output $P_{mpp}$ (Wp) | 250 | Max. DC power (W) | 44,000 |
| Nominal Voltage $V_{mpp}$ (V) | 29.65 | Max. DC Voltage (W) | 1000 |
| Nominal Current $I_{mpp}$ (A) | 8.47 | Start Voltage (V) | 250 |
| Short-Circuit current $I_{SC}$ (A) | 8.80 | PV Voltage Range (V) | 180–1000 |
| Open-Circuit voltage $V_{OC}$ (V) | 37.98 | Nominal voltage (V) | 700 |
| Module Conversion Efficiency (%) | 15.37 | Full load MPP voltage range (V) | 550–800 |
| | | Max. input current (A) | 38 |
| | | Max. input current/ string (A) | 12 |
| | | Max. AC apparent power (kVA) | 40 |
| | | Max. output current (A) | 50 |
| | | Max.efficiency (%) | 98.2 |

Therefore, from the subsystems that make up the entire system;

✓　The eastern group's 34.50 kWp capacity with 24 rows and 138 modules,
✓　The western group's 36.75 kWp capacity with 24 rows and 147 modules,
✓　The northern group's 7 kWp capacity with 4 rows and 28 modules,
✓　The southern group's 6.50 kWp capacity with 4 rows and 26 modules are of 56 rows and 339 modules with a total of 84.75 kWp corresponds to a power plant.

Figure 9 demonstrates the four subsystems specifications which have module data, installation type, shadings, azimuth, orientation, and inclination angles. The properties defined for each subsystem in the planar area facing four different directions belong to that system. Since the areas and facade inclinations are different, the numbers of the panels, the positioning angles, and the shadings are also different in parallel. While every panel is fixed the orientation angle of nearby 180°, the inclination angle of 24°, the azimuth angle is depended on the facade as seen in Figure 9.

In addition to the panels, we have a total of four inverters in this simulation. These are connected to the east-west group two inverters that each power has AC 40 kWp and the north-south group two inverters that each power has AC 8 kWp, with a total of four inverters is integrated into the whole system which is in Figure 10.

The higher produced energy values belong to the west-east group; the fewer are south-north, respectively. The energy produced from each facade group is integrated into the grid depending on four different inverters. Annual production data per inverter is symbolized in the graph below. According to these data, it is seen that the inverter, which is expected to have highest production with the number of panels, is in the western group and the inverter with the least number is in the southern group. The eastern group inverter pursues the western group inverter regarding energy production. The energy production of the northern and southern group inverters is very close to each other can be explained as follows; the fact that the northern group has added two panels more power requires a more significant advantage, while the southern group shows a higher performance by its position.

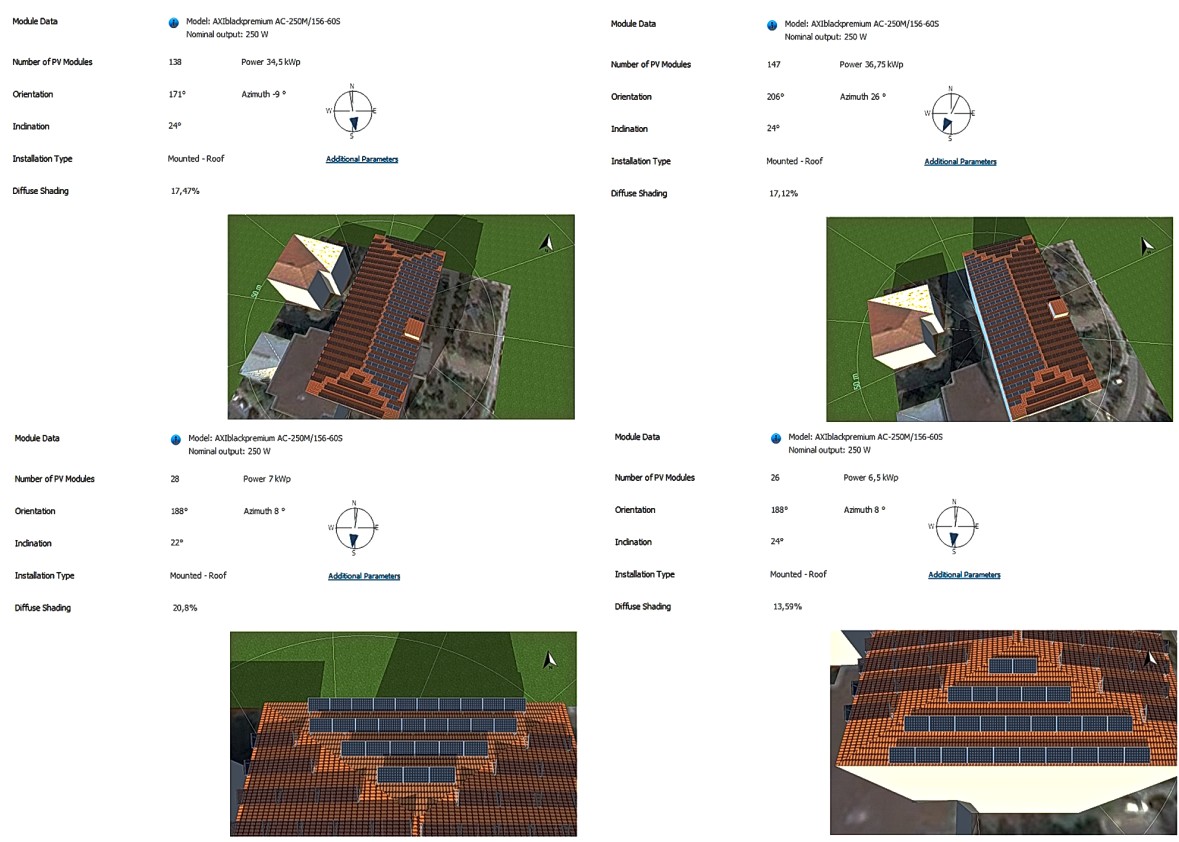

**Figure 9.** Four subsystem the east-west-north-south groups of PV modules.

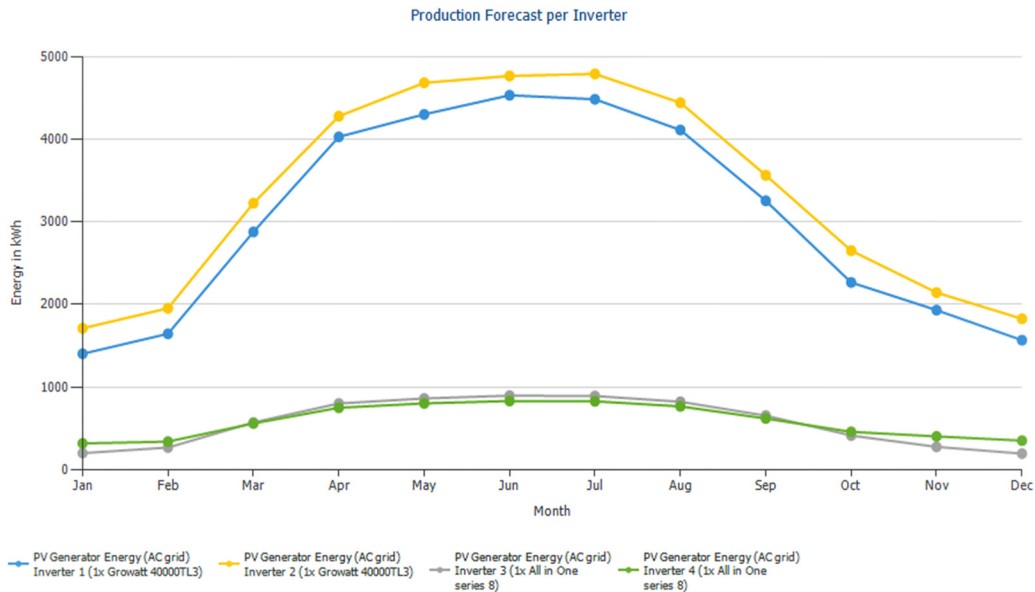

**Figure 10.** Inverter group outputs.

When selecting cable type, cross-sections, and its components such as circuit breaker and residual current devices, different way of the optimum simulation which has the best result was tried many times. Finally, selected cable specifications and components that exist in the simulation are shown in Table 6.

**Table 6.** Cable Overview.

| Cable Type | Cross Section | Material | Total Quantity | Components Type | Quantity |
|---|---|---|---|---|---|
| AC Cable | 10 mm$^2$ | Copper | 24 m | B80 A, Circuit Breaker | 2 |
| AC Cable | 16 mm$^2$ | Copper | 24 m | 80 A/300 Ma, FI/RCD | 2 |
| AC Cable | 2.5 mm$^2$ | Copper | 48 m | B20 A, Circuit Breaker | 2 |
| DC Cable | 2.5 mm$^2$ | Copper | 280 m | 20 A/100 Ma, FI/RCD | 2 |

The distance of the cable from panels to the distribution point was known in advance. Therefore, the cable cross-sections can be calculated and selected the components via entered the distance of the cables.

The appropriate cable cross-sections, breakers, and residual current devices for each PV panel groups are selected according to the optimum voltage drop calculation as seen in Figure 11. Performing all the inputs, the PV*SOL produces distinctive results for discrete conditions of the PV system. These results are affiliated with the performance ratio, the performance of the system, annual production forecast, etc. Besides, financial outcomes can be shown in the PV*SOL, which are below in Table 7.

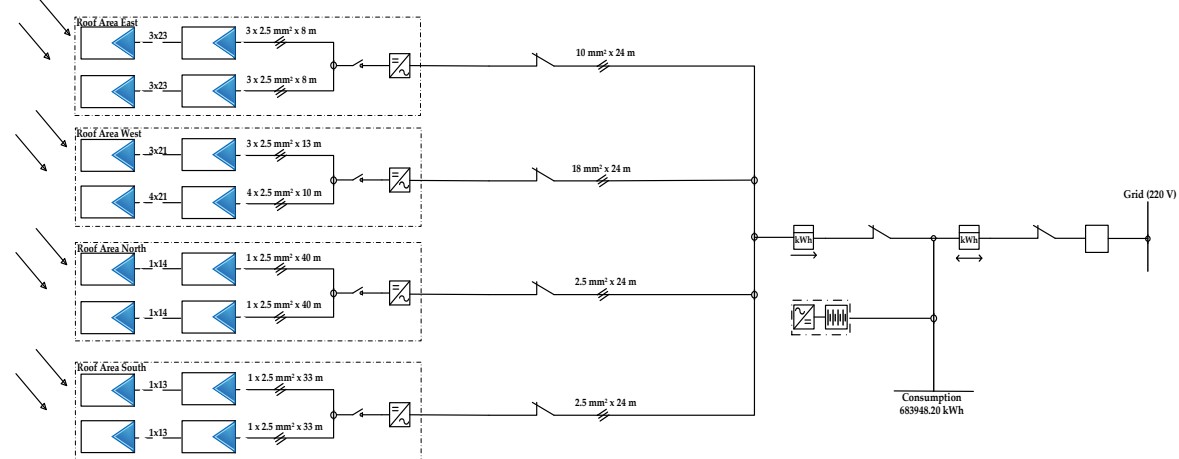

**Figure 11.** System array model.

**Table 7.** The PV*SOL Output Results.

| Results | | Type |
|---|---|---|
| PV Generator Output | 84.75 | kWp |
| PV Produced Energy | 90,298 | kWh/year |
| Specific Annual Yield | 1065.46 | kWh/kWp |
| Performance Ratio | 72.9 | % |
| Avoided $CO_2$ Emission | 54.179 | kg/year |
| Energy Loss (Shading) | 3.7 | %/year |

Thanks to the simulation in the PV*SOL, the above results were acquired as output of the system. If we look over the results obtained from the simulation as shown in Table 7, the PV power plant likely to be installed with produced energy, carbon dioxide emissions, and performance ratio has been turning out promising outputs.

Approximately 13.2% of the 683,948.201 kWh energy consumed in the whole year is provided from the generated PV solar energy that is 90,298 kWh. This situation is depicted in Figure 12. The red line indicated in Figure 12 refers to the energy consumption data. The green one is energy data that can be generated by PV panels according to the simulation results. Established by the PV panels, the energy consumption data which will be fed from the utility grid is understood to be the blue one.

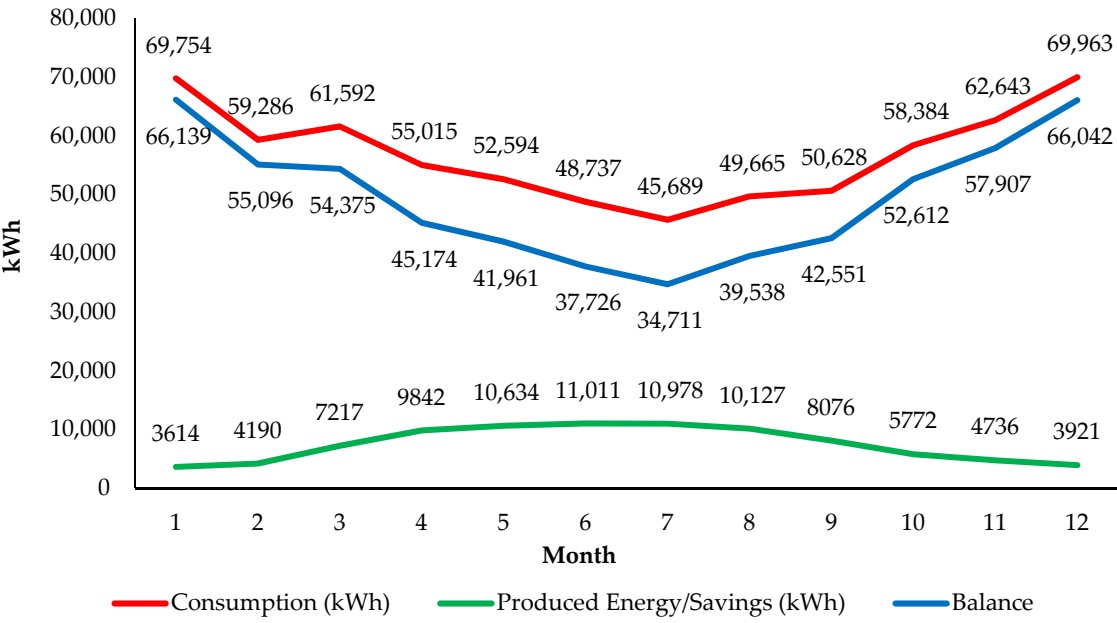

**Figure 12.** Monthly energy balance.

As known in the Northern Hemisphere, while solar energy which is produced in summer months is high, it is low due to inadequate sunbathing and solar irradiance level in the winter months. On the other hand, the electricity consumption of the faculty building especially in the winter months, reaches the highest values in December and January, whereas it is lower in the summer months due to the summer holiday.

Moreover, the AC output of the relevant system can be expressed with annual hourly data in the simulation, in Figure 13.

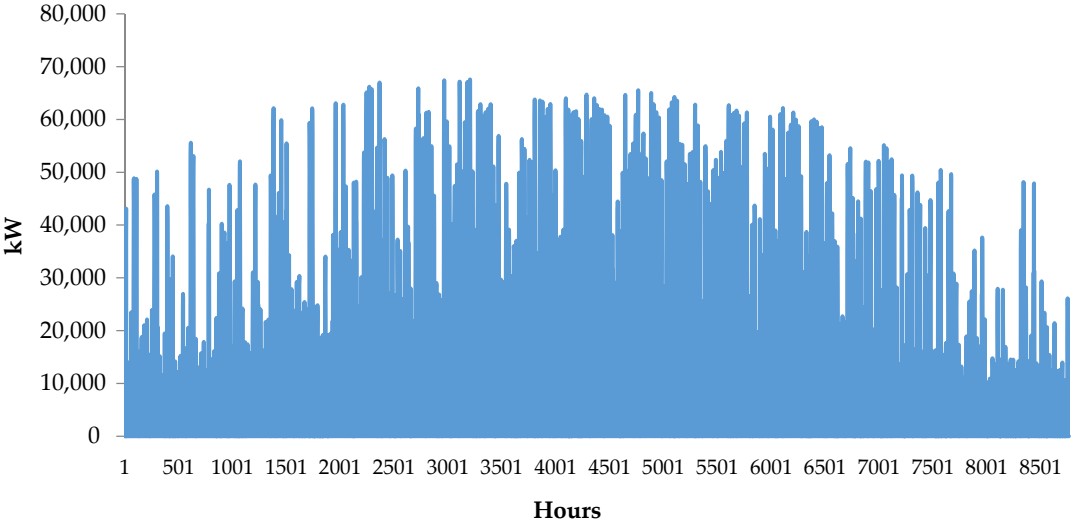

**Figure 13.** Alternating Current (AC) output of the system.

Financial analysis is crucial for all study of the simulation. PV system analyses without performing financial calculations are not sufficient. Therefore, carrying out a simulation comprehensively, this data also is required to complete and acquire real results. Some economic parameters are categorized in Table 8. According to the simulation results, 5000 Turkish Lira (₺) per kWp has been foreseen. Since the installed power of the system is 84.75 kWp, a total cost of 423,750 ₺ will be spent to establish the system.

**Table 8.** The PV*SOL Output Results.

| Economic Results | Unit | |
|---|---|---|
| Cash Flow | 423,750 | ₺ |
| Return on Assets | 13.2 | % |
| Electricity Production Costs | 5000 | ₺/kWh |
| Amortization Period | 12.52 | years |
| Annual Saving Value | 42,316.8 (7296) | ₺ ($) |

In accordance with the simulation outputs, 13.2% of the annual energy procurement will be supplied from the PV power plant to be installed. The payback period for the entire system, which is calculated based on the inflation rate of the annual energy costs, is found to be approximately 12.52 years. Owing to the cost of installation is high, and the duration of the payback period has been increasingly extended.

Self-financing regarding the total of investments, one-off payments, minus subsidies, has been affecting positively the economic output results such as return on asset and amortization period. Through these output results, the system designer can predict their plan without tentative, also design other systems near intrinsic accuracy. Thus, the simulation projects which have notable results may need subsidies of the government in order to make it as a real application project.

Another simulation tool called "PVWatts®" has been as well used in calculations [37] to be able to compare and verify the PV*SOL simulation. Thus, the correctness of the results was tested by this tool. This web-based simulation tool is developed by National Renewable Energy Laboratory (NREL), which estimates the power generation for the roof-mounted PV system. The prediction for the value of this energy is the product of the AC energy and the average retail electricity rate.

The results of PVWatts® indicate that total potential AC system output is nearly 106,776.33 kWh/year shown in Figure 14 and total annual saving value as shown in detail in Figure 15 is $ 7875 (45,675 ₺) regarding giving guidance and ideas before the simulation to be done in the PV*SOL.

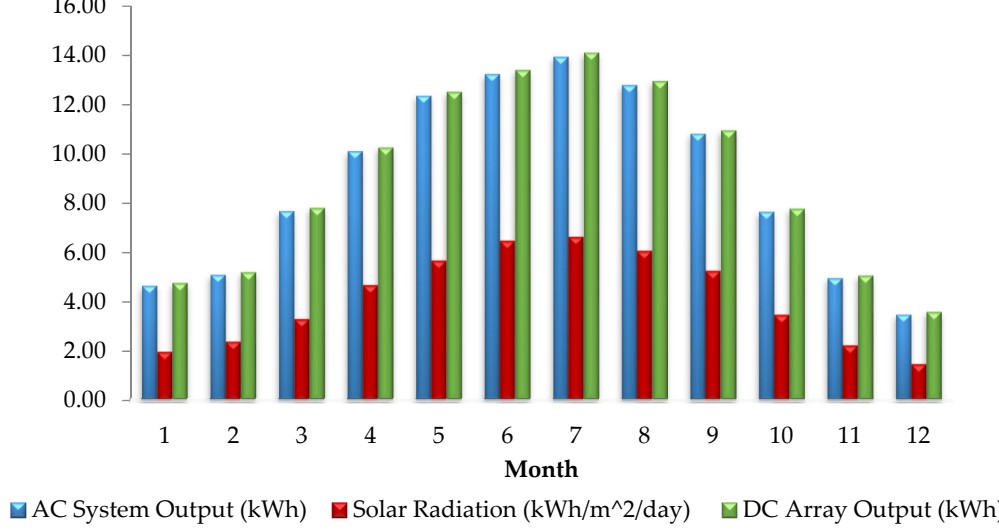

**Figure 14.** The annual the PVWatts® data.

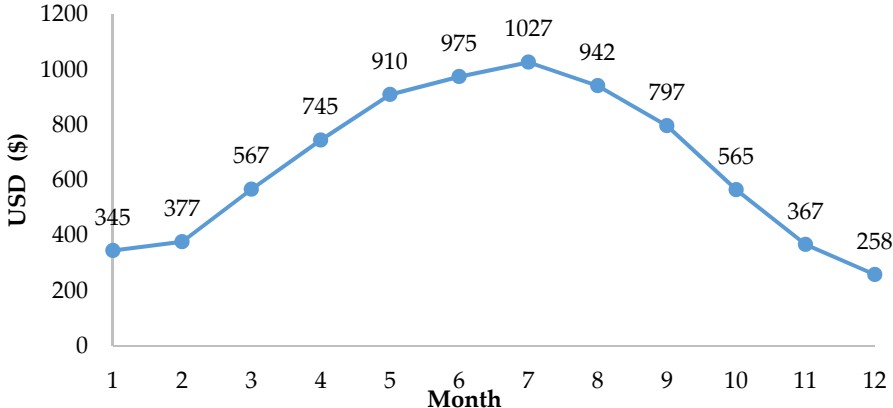

**Figure 15.** Monthly average saving values of the PVWatts® simulation.

## 5. Discussion and Conclusions

The PV*SOL simulation of the RSPS is designed for the faculty building, has a capacity of 84.75 kW. As a result, of the simulation, annual energy production of the system is 90,298 kWh. When the electricity fee per kWh is calculated as $0.0808, a total of $7296 (42,316.8 ₺) is saved via the PV*SOL. It has been estimated that the mounted RSPS will provide 13.2% of the Faculty of Technology's annual electrical energy consumption. According to the factors being 4.18 kWh/m$^2$/day solar irradiation and 7.5 daylight hours respectively for the location, the average performance ratio of the system is 72.9% with an annual specific yield of 1065.46 kWh/kWp, shows that annual $CO_2$ emission can be avoided as 54.179 kg/year, has noteworthy results.

Electricity production cost is about 5000 ₺/kWh, so the payback period for the given system is almost 12.52 years. Although in the PV*SOL simulation the annual PV produced energy is 90,298 kWh, the calculations made with PVWatts result in 106,776.33 kWh. Correspondingly, the annual savings value is $7296, compared to $7875 for PVWatts. Furthermore, the *SPR* of manual calculations is higher than the PV*SOL calculations (69.04% < 72.9%).

For the *SPR* value in manual calculation, not all of system efficiencies and shading effect are taken into account. These numerical differences indicate that the PV*SOL software, which many variables account for, produces closer and more logical results, close to reality.

The use of RSPS in university buildings is essential, not only in terms of increasing the use of renewable energy but also in the sustainability of energy. If the simulation study becomes a practice, students can learn to see these systems and increase awareness of the use of renewable energy in society. The results are exemplary for other faculty buildings and may be caused by the evaluation of the faculty building in the green building category.

This simulation study and analysis may be considered to be a feasibility study for the real system, which is likely to be projected. The real system, which is intended to be transformed into a project, is of importance for initiating transitioning to smart grid technology with a DG infrastructure. As a future study, we plan to begin an implementation project in which national or international support will be included.

**Author Contributions:** Conceptualization, A.N.A. and E.D.; methodology, A.N.A.; software, A.N.A.; validation, E.D., A.E.K. and A.F.B.; formal analysis, A.N.A.; investigation, A.N.A. and E.D.; resources, A.N.A., A.E.K. and A.F.B.; data curation, A.N.A. and E.D.; writing—original draft preparation, A.N.A.; writing—review and editing, E.D., Y.Y. and F.B.; supervision, E.D., Y.Y., and F.B.

**Funding:** This research received no external funding.

**Acknowledgments:** Special thanks to Marmara University-BAPKO (Bilimsel Araştırma Projeleri Birimi) with the project whose number is FEN-E-120314-0069 for their support.

**Conflicts of Interest:** The authors declare no conflict of interest.

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
