# Peer review of "Performance Analysis of a Grid-Connected Rooftop Solar Photovoltaic System"

_electronics, doi:10.3390/electronics8080905_

Round 1
Reviewer 1 Report
This article is well written and begins with an extensive literature review. All equations are well explained and presented. The paper describes a fairly straightforward piece of research using standard software but calculation of PV output on university buildings will be of interest to academics and students. In addition, the work has been verified with a second software package.
Specific Comments:
Where does the information in Table 3 (Potential of The RSPS in Turkey) come from?
Line 292: “characteristic overcurrent length (Ichar) can be calculated as” – something is missing here.
Line 441: “being calm of the faculty building”. Please reword this phrase, the meaning is not clear in this context.
Line 442: “when sunbathing”. Please reword this phrase, the meaning is not clear in this context.
Author Response
Response to Reviewer 1 Comments
At first, we would like to extend our special thanks to you to contribute to our paper by reviewing with your valuable comments.
Comments and Answers
This article is well written and begins with an extensive literature review. All equations are well explained and presented. The paper describes a fairly straightforward piece of research using standard software but calculation of PV output on university buildings will be of interest to academics and students. In addition, the work has been verified with a second software package.
Specific Comments:
Point 1:
Where does the information in Table 3 (Potential of The RSPS in Turkey) come from?
Response 1:
Thank you for your attention, the information of the table was referred to as [23].
Point 2:
Line 292: “characteristic overcurrent length (Ichar) can be calculated as” – something is missing here.
Response 2:
Yes, there is a recurring text. The paragraph had been written two times inadvertently, was corrected with your help.
Point 3:
Line 441: “being calm of the faculty building”. Please reword this phrase, the meaning is not clear in this context. Line 442: “when sunbathing”. Please reword this phrase, the meaning is not clear in this context.
Response 3:
As you mentioned in your comment, these sentences were edited as below;
“As known in the Northern Hemisphere, while solar energy which is produced in summer months is high, it is low by reason of inadequate sunbathing and solar irradiance level in the winter months.
However, the electricity consumption of the faculty building especially in the winter months, reaches the highest values in December and January, whereas it is lower in the summer months due to the summer holiday.”
Best Regards
Reviewer 2 Report
The subject of the article is currently very much needed - it concerns the use of currently popular PV technology, but below I post a few questions and comments about the article.
Comments, questions and doubts:
Line 18 - What did the authors mean by "remarkable power"?
The introduction is quite long and detailed - this is an advantage.
In figure 1 I do not fully understand the entry on the axis "1000 tons of oil equivalent" - please explain.
The value 1013.6 in the graph of figure 1 is incorrect.
Lines 106-108 - The sentence does not sound quite right (in Table 1). The trend mentioned in the sentence can not be seen in this table.
Where do the data from table 3 (source) come from and what year are they from?
Line 166 - The abbreviation RES is not explained.
Equations from 6 to 11 etc. - what are the different symbols "x" from "*". Why are both forms used? The most correct is probably a "small dot", which does not occur here at all.
Generally, I miss the literature sources used in the article of equations and catalog sheets, eg PV panels and inverter. By default, the equations given without sources should be property of the authors, and they are not.
Line 256 - SPR value = 96.04% seems too high, please check it out. Values of the order of 70-85% are more real.
Lines 276-277 - The symbols from equation 13 should be explained / given in the text.
Line 292 - unfinished sentence?
Patterns 21, 22, 24, 29, 30 - I personally do not like long parameter names, I would prefer short symbolic entries.
Is the data from Figure 6 real or simulated? If real, from what period of time (what years)?
Table 5 - How many inputs does each inverter have?
Line 390 - Specify the drawing number.
Lines 400-402 - Have the same inverters been selected for all panels groups? That's what I understand from the description. For groups below 8 kW, it is not worth choosing inverters of lower power? I am asking for a substantive answer.
The inscriptions in the diagram in Fig. 11 are hardly visible, similarly in Fig. 9. I suggest to enlarge the font if possible.
How have cable cross-sections been selected or calculated?
Have you wondered how you could reduce the "amortization period" of this system?
Please, complete the article and answer the above doubts.
Author Response
Response to Reviewer 2 Comments
At first, we would like to extend our special thanks to you to contribute to our paper by reviewing with your valuable comments.
Comments and Answers
The subject of the article is currently very much needed - it concerns the use of currently popular PV technology, but below I post a few questions and comments about the article.
Comments, questions and doubts:
Point 1:
Line 18 - What did the authors mean by "remarkable power"?
The introduction is quite long and detailed - this is an advantage.
In figure 1 I do not fully understand the entry on the axis "1000 tons of oil equivalent" - please explain.
Response 1:
According to the annual data, the whole faculty building has 1899.836 kWh daily average energy consumption value. Also, the faculty includes three labs that spend most of the electricity in the building. These are Control Lab, Renewable Energy Lab, and Electrical Machines Lab which can consume up to 50-kWp power. Thus, the produced 84.75-kWp power can supply these labs consumption with this remarkable power. This is not very high value but prior to beginning to the simulation, we did not hope to arrive at this power that can produce on the roof of the faculty, so this “remarkable power” expression was used as well.
As you know, tonne of oil equivalent (toe) is a unit of energy, defined as the amount of energy released burning one tonne (1000 Kilograms) of crude oil. Thousand tonnes of oil equivalent (ktoe) or 1000 tons of oil equivalent is equal to 11630000 kWh energy. The toe is used to describe large amounts of oil or natural gas in transport or consumption. 1 toe is equal to 41867999999.99929 Joule. The summary of this explaining was added to relevant text.
Point 2:
The value 1013.6 in the graph of figure 1 is incorrect.
Response 3:
Thanks for your caution. It was corrected.
Point 3:
Lines 106-108 - The sentence does not sound quite right (in Table 1). The trend mentioned in the sentence cannot be seen in this table.
Response 3:
According to table, in 2006, there is no used solar energy resources record in Turkey. So, the data according to 2014 does not exist in Table 1. But it is known from reference [23] that the solar capacity is also 40 MW in 2014. With your help, it was added.
Point 4:
Where do the data from table 3 (source) come from and what year are they from?
Response 4:
Thanks for your attention. The reference was edited.
Point 5:
Line 166 - The abbreviation RES is not explained.
Response 5:
The abbreviation of renewable energy source (RES) was given in Line 46 for the first time.
Point 6:
- what are the different symbols "x" from "*". Why are both forms used? The most correct is probably a "small dot", which does not occur here at all.
Response 6:
Of course, they cause a dilemma. It was edited.
Point 7:
By default, the equations given without sources should be property of the authors, and they are not.
Response 7:
There are a lot of equations in the paper. The equations were given references that do not belong to us.
Point 8:
Line 256 - SPR value = 96.04% seems too high, please check it out. Values of the order of 70-85% are more real.
Response 8:
It had been written inadvertently. Instead of 96.04%, it was corrected as 69.04. The calculation exists in point 20.
Point 9:
Lines 276-277 - The symbols from equation 13 should be explained / given in the text.
Response 9:
It was edited and referred in the text.
Point 10:
Line 292 - unfinished sentence?
Response 10:
There is a recurring text. The paragraph had been written two times inadvertently, was corrected.
Point 11:
Patterns 21, 22, 24, 29, 30 - I personally do not like long parameter names, I would prefer short symbolic entries.
Response 11:
They were edited via their abbreviations.
Point 12:
Is the data from Figure 6 real or simulated? If real, from what period of time (what years)?
Response 12:
The data in Figure 6 is real data, we collected this data from the technical office of the University. The time period of the data was mentioned as "the annual average energy consumption of the faculty building as to the 2017 year."
Point 13:
Table 5 - How many inputs does each inverter have?
Response 13:
We have total four inverters in this simulation. These are connected to the Eastern-Western group two inverters that each power has AC 40 kWp and the Northern-Southern group two inverters that each power has AC 8 kWp, with a total of four inverters is integrated into the whole system which is in Figure 10.
They are connected each other as follows
--the eastern group’s 34.50 kWp capacity with 24 rows and 138 modules à AC 40 kWp inverter
--the western group’s 36.75 kWp capacity with 24 rows and 147 modules à AC 40 kWp inverter
--the northern group’s 7 kWp capacity with 4 rows and 28 modules, à AC 8 kWp inverter
--the southern group’s 6.50 kWp capacity with 4 rows and 26 modules, à AC 8 kWp inverter
Point 14:
Line 390 - Specify the drawing number.
Response 14:
It was edited as Figure 9.
Point 15:
Lines 400-402 - Have the same inverters been selected for all panels groups? That's what I understand from the description. For groups below 8 kW, it is not worth choosing inverters of lower power? I am asking for a substantive answer.
Response 15:
The simulation tools have some inverter options in its database to select the appropriate inverters. Once the designer passes to select the appropriate inverter for the PV modules, the designer obligates to comply with the simulation suggestions for the optimum installation. Thus, for the northern and southern group (7 kWp and 6.50 kWp) we could choose these inverters which have at least that power.
Point 16:
The inscriptions in the diagram in Fig. 11 are hardly visible, similarly in Fig. 9. I suggest to enlarge the font if possible.
Response 16:
The Figure 9. and 11. was revised again.
Point 17:
How have cable cross-sections been selected or calculated?
Response 17:
Yes, it has been selected at least cost up to the simulation permit. Also, with this fine question, the cable overview and cable components such as breaker and residual current device were added as Table 6.
Point 18:
Have you wondered how you could reduce the "amortization period" of this system?
Response 18:
Yes, in order to reduce the amortization period and ascend the SPR, different simulation types had been tried many times. Establishing the system type, choosing the available location on the roof, selecting the optimum price-performance PV module and other devices, and the other thing that affects the amortization period was chosen carefully. The optimum system as much as we can which has the best result is in the paper.
Point 19:
Please, complete the article and answer the above doubts. found that sections 2 & 3 should be re‐organized and be shortened. It may be easier for the readers if the authors define properly the mixture of regression model and the class‐ membership equation first before moving to the computation of the GINI and of the Polarization of subgroups.
Response 19:
This warning was paid attention, checked and edited.
Point 20: Sections 2.1 and 2.2 are too long and can be significantly reduced. In section 2.1 the authors assume the condition uk > uj, but this does not appear anywhere else in the calculation of the mixture of regression model.
Response 20:
This calculation was written in the relevant place as below;
“According to this simulation results,
Et=90298 kWh, n=339 panels, A=1. 1.643 m2, H=1527 kWh/m2, η=15.37%
with is calculated as 69.04%.”
Point 21:
After equation (10) all the other equations are not numbered.
Response 21:
It was checked
Best Regards
Reviewer 3 Report
The author proposes an interesting topic suitable for this journal. PV source is a great choice towards the sustainability. I think that the quality of this work is very high and I can also accept it. However, I require major revisions because I ask an additional elaboration of idea:
PV plays a key-role from an economic and environmental perspective. This aspect must be clear in section 1. I suggest some works. i) https://www.sciencedirect.com/science/article/pii/S0959652618306280 ii) https://www.mdpi.com/2076-0760/7/9/148 What is the methodology used in this work. All technical equations must be associated to a proper source What are the limits of your methodology? All rooftop solar PV systems have the same conditions of work? For example, can substitute dangerous substances favouring also an environmental challenge. What is the contribution of the share of self-consumption? What is the role of subsidies? What is your final message? FInal provide it in Line 23-25 (the current state is not adequate)I think that the following step will be an accept ;)
Author Response
Response to Reviewer 3 Comments
At first, we would like to extend our special thanks to you to contribute to our paper by reviewing with your valuable comments.
Comments and Answers
The author proposes an interesting topic suitable for this journal. PV source is a great choice towards the sustainability. I think that the quality of this work is very high and I can also accept it. However, I require major revisions because I ask an additional elaboration of idea:
Point 1:
PV plays a key-role from an economic and environmental perspective. This aspect must be clear in section 1. I suggest some works.
https://www.sciencedirect.com/science/article/pii/S0959652618306280 https://www.mdpi.com/2076-0760/7/9/148
Response 1:
Thank you so much for your suggestion. The section I was readjusted and added to the main text as you mentioned;
“The two main resources for very large scale renewable energy (RE) harvesting are the wind and the solar resource. The wind resource may show variability and be limited, but the harvesting of the solar resource is easier and more common than the wind.
So, solar energy is a suitable technology for both small and large scale applications. It is clean energy according to the principle of sustainability. Especially, solar energy is the fastest-growing energy technology across the world. Solar energy includes the two main way as photovoltaic (PV) and concentrating solar power (CSP). PV is unquestionably more applicable than CSP [7, 8].
Therefore, generating energy with solar PV is the most trending application in terms of using RESs across the world. Recently, in developing countries across the world, decisions are being debated and a transitioning to the dependence from fossil fuels to RES. Among renewable energy, especially the PV systems play a vital role in this transition for PV applications.”
Point 2:
What is the methodology used in this work? All technical equations must be associated to a proper source What are the limits of your methodology?
Response 2:
The methodology used in this paper can be expressed as and was inserted into the relevant place.
The methodology of the paper with PV*SOL software supports system designers in deciding the PV system. The software evaluates the necessary data and calculates the solar yield. The orientation and inclination of the PV panel should be defined in advance. After choosing one of the three types of solar PV modules such as monocrystalline, polycrystalline, and thin film, the number of module size of the module array, the software automatically determines the location of the PV system. Alternatively, related to the system location, any place can be selected on the world map. PV*SOL software uses climate data from PV*SOL database.
The simulation contains quite a complicated process. Generally, the main steps of the simulation can be expressed such as selecting the location (the meteorological data), selecting PV modules and their planning, selecting inverters, cables or other devices requirements. After these complex definitions, and detailed design a detailed report can be printed out. With the climate data for the location and the characteristics selected for the solar PV system, the expected annual yield of the system can be calculated using a detailed hourly simulation. The simulation prefers the PV*SOL software calculation model equations on the Valentin Software, which discuss in 3.3. section so as to calculate the whole output results.
Point 3:
All rooftop solar PV systems have the same conditions of work? For example, can substitute dangerous substances favouring also an environmental challenge. What is the contribution of the share of self-consumption? What is the role of subsidies?
Response 3:
RSPSs have nearly same working conditions that change with the location coordinates, the meteorological data, planning the modules, and selected PV modules, inverters, cables or other devices requirements.
These systems also are a part of zero emission systems so they cannot exist as dangerous substances favouring environmental challenge.
Self-financing regarding the total of investments, one-off payments less subsidies, has been affecting positively the economic output results such as return on asset and amortization period. Through these output results, the system designer can predict their plan without tentative, also design other systems near intrinsic accuracy. Thus, the simulation projects which have notable results may need subsidies of the government in order to make it as a real application project.
Point 4:
What is your final message? Final provide it in Line 23-25 (the current state is not adequate)
I think that the following step will be an accept ;)
Response 4:
The abstract part of the paper is limited as only 200 words so we only were able to add a sentence as follows,
“This study can be considered as a basic feasibility report prior to moving on to the implementation project for buildings.”
Best Regards
Round 2
Reviewer 3 Report
Congratulations